# Assessing the bread-leavening ability of wild yeasts isolated from selected fruits collected from local markets

Eshet Lakew Tesfaye[1]*, Anteneh Tesfaye Tefera[2], Diriba Muleta[2]

1 Department of Biotechnology, College of Natural and Computational Sciences, Hawassa University, Hawassa, Ethiopia, 2 Institute of Biotechnology, Addis Ababa University, Addis Ababa, Ethiopia

* eshetbiot@gmail.com

## Abstract

Yeasts play a crucial role in converting sugars in flour into carbon dioxide ($CO_2$) and ethanol, enabling dough leavening. This study focused on identifying efficient wild yeast strains isolated from various fruits for leavening under controlled laboratory conditions. Of the 88 wild yeast isolates obtained, three (AAUGr5, AAUOr7, and AAUPi3) were selected based on their ability to produce $CO_2$ efficiently, using D-glucose (3 g/L) as a substrate and their non-hydrogen sulfide ($H_2S$) production character. These isolates were tentatively identified as members of the *Saccharomyces* genus through cultural, morphological, and biochemical analyses and optimal physicochemical profiling. Optimal growth conditions in Yeast Extract Peptone Dextrose (YPD) medium were a pH of 5 and a temperature of 30 °C. Maximum growth was observed with 2% (w/v) D-glucose and 0.5% (w/v) NaCl concentrations, achieving peak biomass after 96 hrs of incubation. Notably, isolate AAUGr5 displayed superior leavening performance compared to commercial yeast with 9.09% and 8.92% at 30 °C and room temperature, respectively. Furthermore, a co-culture of these $H_2S$-free wild yeast strains demonstrated enhanced leavening activity, underscoring their potential for baking applications to produce high-quality bread.

## 1. Introduction

For millennia, humans have utilized the domesticated *Saccharomyces cerevisiae*, known as baker's yeast, to prepare bread. Foods that have been baked or fermented hold significant cultural and economic value in our society [1]. In bread, beer, wine, and other uses, *Saccharomyces cerevisiae* continues to be the preferred organism [2]. Bread contributes sufficient calories and other elements, including proteins, vitamins B and E, and minerals. It is usually made from a combination of wheat flour, yeast (*Saccharomyces cerevisiae*), salt, and water [3]. A vital function of biological leavening agents is their ability to raise flour dough. Biological leavening agents, such as yeasts and lactic acid bacteria, can use the sugars in flour to produce carbon

**Data availability statement:** All relevant data are within the paper and its Supporting Information files.

**Funding:** The author(s) received no specific funding for this work.

**Competing interests:** The authors have declared that no competing interests exist.

dioxide and other compounds, including various metabolites [4]. The most widely utilized microbe in bread baking is *Saccharomyces cerevisiae*. The primary role of baker's yeast in bread baking is leavening, which is accomplished by generating carbon dioxide through the alcoholic fermentation of sugars. This increases the volume of the dough and gives the bread its distinctively light and spongy texture, as well as a wonderful flavor [5].

To meet global demand commercial baker's yeast has been utilized for centuries to leaven bread. Therefore, the development of the bread business has been greatly aided by yeasts as a starting culture. The dough can be leavened using a variety of naturally present mixtures of wild yeasts, coliform bacteria, saccharolytic Clostridium species, and heterofermentative lactic acid bacteria, however, *Saccharomyces cerevisiae* is the most effective starter [6]. The characteristics that make modern baker's yeast strains the best choice from other yeast strains include maximal growth temperature around room temperature, relative resistance to low pH, and rapid growth in the presence of glucose [7].

In Ethiopia, baked foods are extensively consumed and have a significant economic impact [8]. Ethiopia's bakery industry is expanding steadily as a result of the country's growing population and urbanization trends, as well as the country's increasing reliance on fast food in place of traditional eating habits. These and other factors combined play a significant role in raising the demand for bread, especially bread that is baked commercially [9]. All baking industries in Ethiopia use baking powder, and the country imports most of its baker's yeast, mainly from China [10]. The trend showed that a large amount (385,107.88 kg) of baking powder is imported every year for baking purposes. The country's demand to produce better bread has grown recently. As a result, bakeries are now importing starting cultures worth significantly more than a million US dollars [10], which is detrimental to the Ethiopian economy.

$H_2S$ is one of the unpleasant odors that some strains of *Saccharomyces cerevisiae* produce during dough leavening [11]. Therefore, yeasts intended for use in bread leavening should not produce this undesirable compound [12]. Because *Saccharomyces cerevisiae* is a sugar-loving microbe, it can be isolated from sources high in sugar. Microorganisms use fruits as part of their phyllosphere habitat [13]. Fruits serve as a common food supply for numerous yeast species, much as sugar-rich ambient samples [14]. Yeast species utilize the abundant sugar found in fruits and can be isolated from these sources. Many substrates are available for the isolation of yeast species, such as molasses, citrus juice, sugarcane juice, teff dough, wheat dough, shamita, tej, fermented foods, and fermented beverages [9,15]

In the current study, we further investigated the leavening ability of wild yeasts isolated from different fruit resources and examined them for their potential commercial-scale production. We believe this research could reduce Ethiopia's reliance on imports of baker's yeast and help save the economy. Therefore, the study aimed to isolate, select, and assess the ability of wild yeasts from different fruits for dough leavening and their utilization in the bakery industry.

## 2. Materials and methods

### 2.1. Sampling site and sample collection

Fruit samples, eight in each type of fruit, including avocado (*Persea americana*), banana (*Musa acuminate*), grape (*Vitis vinifera*), mango (*Mangifera indica*), orange (*Citrus sinensis*), papaya (*Carica papaya*), and pineapple (*Ananas comosus*), were collected from the two main local marketplaces in Addis Ababa City, Merkato and Atikilit Tera.

### 2.2. Sample preparation

Fruit sample preparation for wild yeast isolation was done following the method indicated by [16] with slight modification. Each fruit sample (100 g) was surface-washed with distilled water, chopped, and crushed with a sterile mortar and pestle. It was then added with sterile distilled water (50 mL) into autoclaved beakers. The fruit homogenate was stored at room temperature for three days to promote fermentation and growth. This study used active dry baker's yeast (Saf instant) from DSM bakery ingredients, Holland, as a standard strain.

### 2.3. Isolation of wild yeasts

Each independently fermented fruit sample (1 mL) was serially diluted from $10^{-1}$ to $10^{-6}$ by transferring into tubes that contained 9 mL sterile peptone water in a test tube under a laminar airflow hood. Aliquots of (0.1 µL) were spread-plated on pre-dried potato dextrose agar (PDA, HIMEDIA) plates containing chloramphenicol (0.1 g/L) to prevent bacterial growth. All plates having the cultures were incubated in triplicate at 30 °C for 48 hrs. Representative colonies from the cultures were taken and purified with a similar, freshly prepared PDA medium. Finally, the purified isolates were transferred to PDA slants and preserved at 4 °C in the refrigerator for further study.

### 2.4. Screening of yeast isolates based on $CO_2$ and $H_2S$ production

Each purified isolate found in this study was inoculated with 5 mL of YPD broth using a Durham tube to measure the amount of $CO_2$ produced by each wild yeast isolate. D-glucose (3 g/L) was added in all tubes and incubated at 30 °C for 24 hrs, and those isolates that produced more amount of $CO_2$ by displacing the medium in the Durham tube were selected.

After obtaining those potential isolates, the production of $H_2S$ by each purified yeast isolate was confirmed by streaking each yeast isolate on plates containing both Kligler Iron Agar (KIA) and Bismuth Sulfite Agar (BSA). The plates containing the two media were incubated at 30 °C for 3 days following the procedures [17]. Black-colored yeast isolates on BSA plates and any blackening of the KIA through the butt or along the inoculation line were used to indicate the formation of $H_2S$ [17].

### 2.5. Identification of selected wild yeast isolates from fruits

**2.5.1. Cultural identification of the yeast isolates.** Cultural characteristics of the yeast isolates were done with slight modification of the procedures in [18]. The purified wild yeast isolates were identified based on visual examination for their growth, color, colony size, and texture on PDA media.

**2.5.2. Microscopical identification of the yeast isolates.** The microscopical identification of yeast isolates was done following the method indicated by [18]. On a glass slide, purified yeast colonies were mixed with a drop of distilled water and spread until the smear dried off. After that, the smear was stained with lactophenol cotton blue and examined under a microscope at a magnification of 100×.

**2.5.3. Biochemical characterization.** Biochemical characterization of the selected yeast isolates was done following the procedures of [19] with modification. The ability of each of the selected yeast isolates to produce $CO_2$ and use D-glucose, fructose, maltose, galactose, lactose, and sucrose as the only carbon source was assessed. Yeast extract

(10 g) and peptone (10 g) were added to distilled water (1000 mL), and the mixture was well mixed. After adjusting the pH to 5 using 1N HCl or NaOH, the medium was brought to a boil. Bromocresol purple carbohydrate (2%, w/v) fermentation indicator was added to an already boiled yeast extract peptone broth (5 mL) amount into screw-capped test tubes containing inversely placed Durham tubes. Similarly, $KNO_3$ (10 g/L) and $(NH_4)_2SO_4$ (10 g/L) were added and used as nitrogen sources. The solutions were sterilized at 121 °C for 15 minutes in separate flasks.

To conduct the carbohydrate fermentation test, yeast cells (6.6 log cfu/mL) from each isolate were inoculated into individual tubes containing 5 mL of yeast extract peptone broth that included 0.5 mL of sterile sugar and nitrogen solution. The inoculated tubes were incubated at 30 °C for 48 hrs in triplicate. The change of color from violet to yellow due to acid production and accumulation of gas bubbles in the inverted Durham tube ($CO_2$ gas production) was taken as a positive result of sugar fermentation. No color change was taken as a negative result.

**2.5.4. Effect of temperature, pH, glucose, and sodium chloride concentrations on the growth of yeasts.** Each selected yeast isolate was grown at different temperature values (25, 30, 35, and 42 °C) by inoculating each actively growing yeast isolate (log 6.6 cfu/mL) into a sterile YPD broth of 50 mL (pH adjusted to 5 before autoclaving) in a separate flask in triplicate.

Flasks were incubated at respective temperature values, and the growth of each isolate was determined by reading the optical density at 550 nm using a spectrophotometer (6405 UV/Vis, JENWAY, United Kingdom) at intervals of 24, 48, 72, 96, and 120 hrs.

To determine the optimum growth pH for the three selected yeast isolates, 1N HCl or NaOH was used to adjust the pH of the YPD broth in separate flasks to 4, 5, and 6. Into a flask with a given pH value, each actively growing yeast isolate (log6.6 cfu/mL) was inoculated and incubated at 30 °C in triplicate. A spectrophotometer (6405 UV/Vis, JENWAY, United Kingdom) at 550 nm was used to read the growth of the yeast isolates at a given pH value at 24, 48, 72, 96, and 120 hrs.

Different D-glucose concentrations (2, 3, and 4%) were prepared to evaluate the ability of the selected yeast isolate to grow in a YPD medium. The pH of the broth medium was adjusted to 5 using 1N HCl or NaOH. In the same way, separate flasks (50 mL) were inoculated with log6.6 cfu/mL of each of the isolates that were actively growing isolates, and flasks were incubated at 30 °C in triplicate. The growth of the yeast isolates was measured at 96 hrs using a spectrophotometer (6405 UV/Vis, JENWAY, United Kingdom) set at 550 nm.

The selected yeast isolates were grown in a 50 mL YPD medium at various NaCl concentrations (0.5, 1, and 1.5%, w/v) at pH 5. Separate flasks were inoculated with each actively growing isolate (log6.6 cfu/mL) and incubated at 30 °C in triplicate. A spectrophotometer (6405 UV/Vis, JENWAY, United Kingdom) was used to determine the growth of the selected yeast isolate after 96 hrs of incubation.

## 2.6. Determination of the leavening ability of the selected yeast isolates

The fermenting ability of selected yeast isolates to leaven bread dough was determined with little modification of the protocol of [9]. Each yeast isolate was grown separately and in co-culture at 30 °C in a flask containing 250 mL YPD broth for 72 hrs. Subsequently, yeast biomass was obtained by centrifuging the culture at 3000 rpm for 10 minutes. The dough was prepared by adding wheat flour (100 g), table salt (0.5 g), table sugar (2 g), sterile distilled water (90 mL), and 1 g yeast isolate (log6.6 cfu/mL) in a 1000 mL measuring cylinder. Commercial baker's yeast (SFI) was used as a control. Similarly, another batch of dough formulations, prepared without yeast isolates, served as the negative control.

The fermenting ability of each yeast isolate was examined individually and in different combinations as co-cultures. The formulations were designated as follows: AAUPi3 (starter 1), AAUGr5 (starter 2), AAUOr7 (starter 3), AAUOr7 + AAUPi3 (starter 4), AAUGr5 + AAUPi3 (starter 5), AAUGr5 + AAUOr7 (starter 6), and AAUGr5 + AAUPi3 + AAUOr7 (starter 7), SFI (starter 8), and no yeast. Each container of fermenting dough with a specific starter culture of log6.6 cfu/mL was incubated at room temperature and 30 °C for 0–8 hrs in triplicate. The volume increment of the fermented dough was measured using a graduated cylinder at 0 hrs and 2 hrs intervals during fermentation for 8 hrs.

## 2.7. Data analysis

The data were interpreted using one-way analysis of variance (ANOVA), and any significant differences were found using post hoc analysis (Tukey's test). SPSS version 20.0 was used to analyze the ANOVA data at the $p < 0.05\%$ significant level.

## 3. Results

### 3.1. Cultural characteristics of yeast isolates

In this study, 88 yeast isolates were recovered from local fruits, purified, and further identified. The purified yeast isolates obtained from avocado (13, 14.8%), banana (14, 15.9%), grape (11, 12.5%), mango (11, 12.5%), orange (15, 17%), papaya (14, 15.9%), and pineapple (10, 11.4%). The cultural characteristics of isolates were observed on PDA as rounded and uneven in shape, with a creamy and white creamy color. The isolates also had both rough and smooth edges (Data not shown).

### 3.2. Production of $CO_2$ and $H_2S$

Out of the 88 isolates of wild yeast, six of the isolates of yeast were selected based on the production of more $CO_2$ observed in the Durham tube within 24 hrs. Of the six yeast isolates that produced higher levels of $CO_2$, isolates AAUMa4, AAUBa2, AAUMa6, and commercial yeast (SFI) were shown to produce $H_2S$ gas. The yeast isolates AAUGr5, AAUPi3, and AAUOr7 were found not to produce $H_2S$ on either of the two media, Bismuth Sulfite Agar (BSA) or Kligler Iron Agar (KIA) (Fig 1 a and b). As a result, AAUGr5, AAUPi3, and AAUOr7 were selected for further analysis.

### 3.3. Morphological characteristics of selected yeast isolates

Under a microscope, AAUOr7 was shown to have a spherical shape, whereas AAUGr5, AAUPi3, and SFI (commercial yeast) had oval shapes. Additionally, AAUOr7 and AAUGr5, two yeast isolates, showed a budding pattern with single, paired, and triplet characteristics. On the other hand, AAUPi3 only revealed a single budding pattern, while SFI showed both single and paired budding patterns (Data not shown).

### 3.4. Biochemical characteristics of selected yeast isolates

Except for lactose, the control and the selected yeast isolates were seen to thrive on all sugars. When given $(NH_4)_2SO_4$ as a nitrogen source instead of $KNO_3$, all isolates and the control group demonstrated growth (Table 1).

### 3.5. Effect of temperature, pH, D-glucose, and NaCl

At different incubation temperatures, the biomass production of the isolates varied from 0.004 to 1.53 OD (Fig 2). The biomass of each yeast isolate was lower at 42 °C, while moderate growth was observed at 25 °C and 35 °C. The maximum biomass was achieved when all yeast isolates were grown at 30 °C. The OD values for each isolate were as follows: 1.53 (AAUPi3), 1.50 (AAUOr7), 1.07 (AAUGr5), and 1.14 (SFI) after 96 hrs of incubation (Fig 2). Extending the incubation period did not result in any increase in biomass at all temperature values (Fig 2).

Fig 3 showed that yeast isolates were able to grow at different pH values (4, 5, and 6) at 30 °C. All of the yeast isolates showed the highest optical density (OD) reading for growth at 550 nm at pH 5. The OD values for the isolate at pH 5 were found to range from 0.95 (AAUPi3) to 0.88 (AAUGr5), whereas 0.99 for (SFI). Our result indicated that both SFI (0.99) and AAUPi3 (0.95) produced more biomass than the other two yeast isolates. It is worth noting that pH 5 was the optimal pH value for the growth of each yeast isolate.

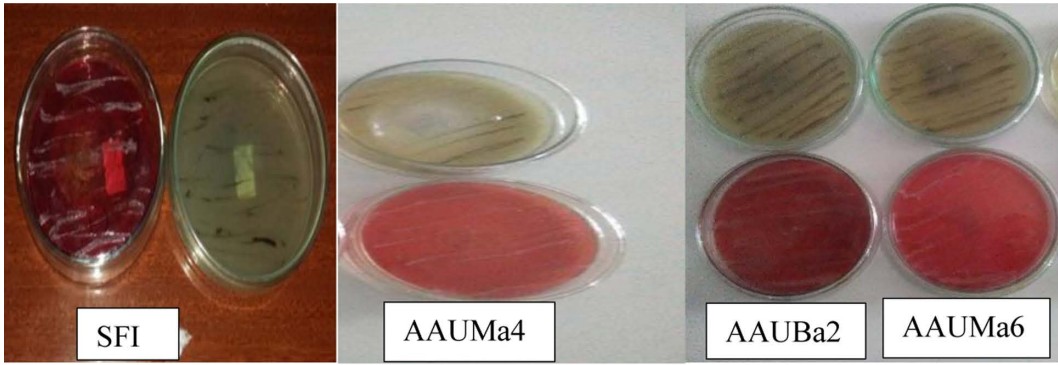

(a)

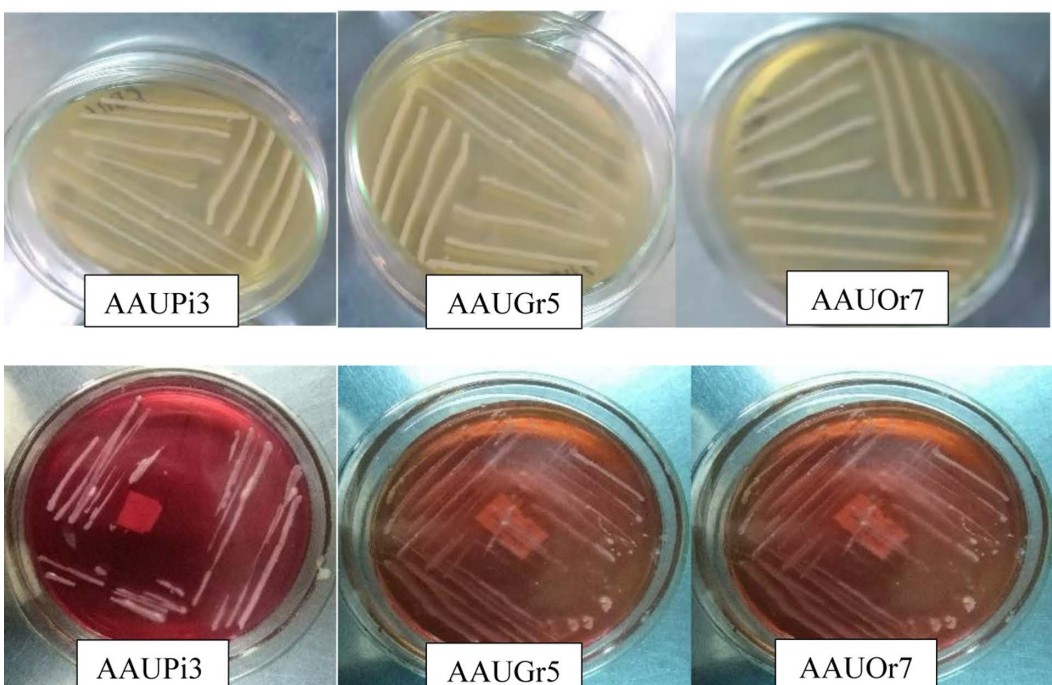

(b)

**Fig 1. Observation of H₂S gas production by culture on BSA (upper) and KIA media (lower).** (a) SFI = commercial yeast, AAUMa4 and AAUMa6 = Addis Ababa University mango, and AAUBa2 = Addis Ababa University banana; (b) AAUPi3 = Addis Ababa University pineapple, AAUGr5 = Addis Ababa University grape and AAUOr7 = Addis Ababa University orange; a = H₂S producers; b = Non H₂S producers.

The growth of each yeast isolate was shown to decline progressively in a YPD medium with 3% and 4% D-glucose. The maximum mean biomass values of the yeast isolates range from 1.81 (AAUPi3) to 1.70 (AAUGr5) and 1.78 for SFI at 2% D-glucose. Similarly, in a medium containing yeast extract and peptone (5 g/L) at 0.5% NaCl (w/v), the biomass of all yeast isolates was at its greatest. Biomass gradually decreased in 1% NaCl (w/v) and further in 1.5% NaCl (w/v). OD reading of the biomass of the yeast isolates was shown to vary between 1.73 (SFI) to 0.97 (AAUOr7) (Table 2).

**Table 1. Biochemical characteristics of selected yeast isolates.**

| Isolate | D-glucose | Fructose | Maltose | Galactose | Lactose | Sucrose | $KNO_3$ | $(NH_4)_2 SO_4$ | Tentative identity |
|---------|-----------|----------|---------|-----------|---------|---------|---------|------------------|--------------------|
| SFI | + | + | + | + | – | + | – | + | *S. cerevisiae* |
| AAUGr5 | + | + | + | + | – | + | – | + | *Saccharomyces* |
| AAUOr7 | + | + | + | + | – | + | – | + | *Saccharomyces* |
| AAUPi3 | + | + | + | + | – | + | – | + | *Saccharomyces* |

AAUGr5 = Addis Ababa University grape, AAUOr7 = Addis Ababa University orange, AAUPi3 = Addis Ababa University pineapple, and SFI = commercial yeast. + = isolate fermented carbon source and utilized the nitrogen source, - = the isolates didn't ferment the carbon source and didn't utilize the nitrogen source.

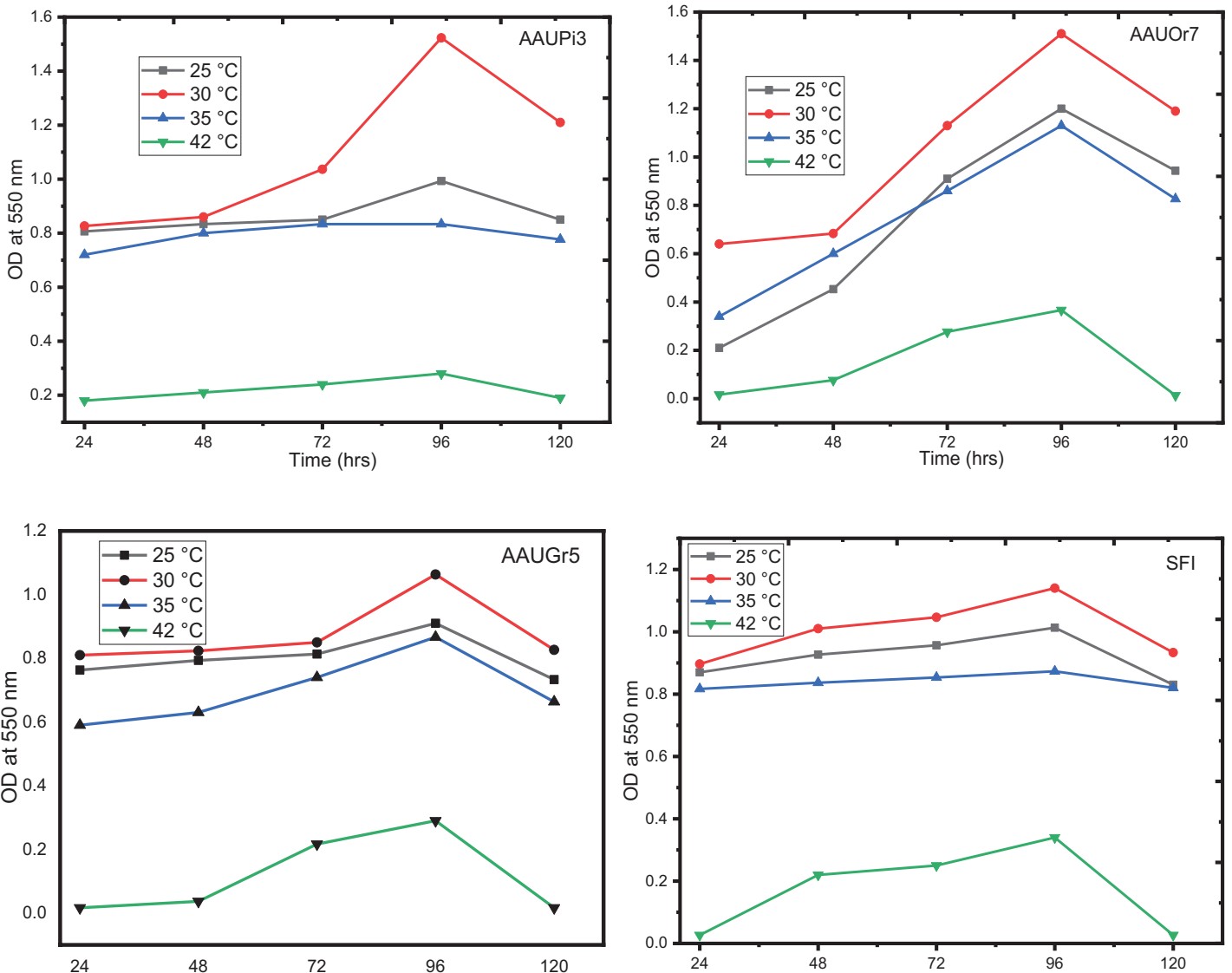

**Fig 2. Effect of temperature on wild yeast growth in YPD in a 24 hrs interval incubation.** AAUPi3 = Addis Ababa University pineapple, AAUOr7 = Addis Ababa University orange, AAUGr5 = Addis Ababa University grape and SFI = commercial yeast.

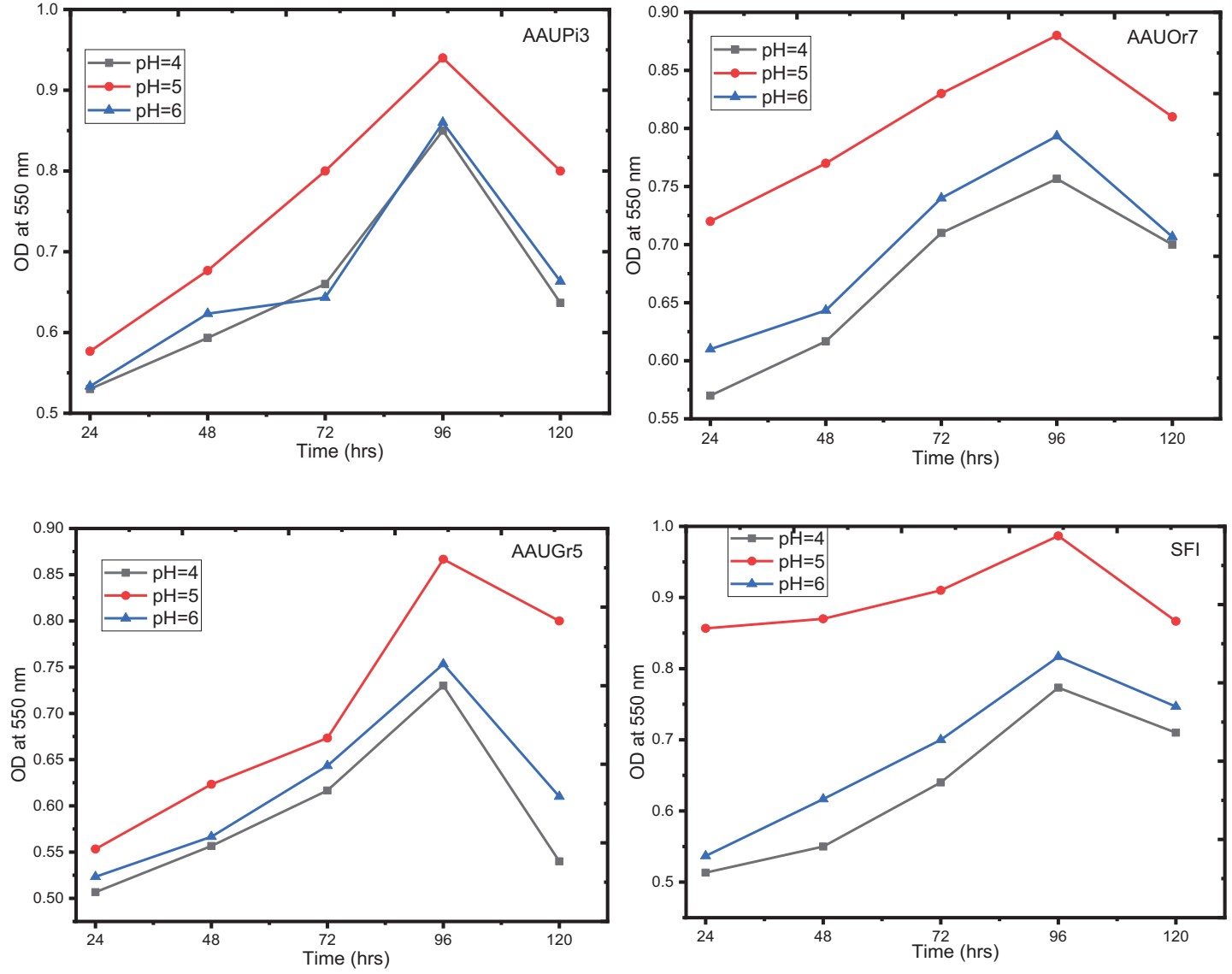

**Fig 3. Effect of pH on wild yeast growth in YPD in a 24 hrs interval incubation time.** AAUPi3 = Addis Ababa University pineapple, AAUOr7 = Addis Ababa University orange, AAUGr5 = Addis Ababa University grape and SFI = commercial yeast.

**Table 2. Effect of D-glucose and NaCl on the aerobic growth of isolated yeast.**

| D-glucose (%) | Growth (OD at 550 nm) of the isolates at different D-glucose concentrations | | | | NaCl (%) | Growth of the isolates (OD at 550 nm) at varying NaCl concentrations | | | |
|---|---|---|---|---|---|---|---|---|---|
| | AAUPi3 | AAUOr7 | AAUGr5 | SFI | | AAUPi3 | AAUOr7 | AAUGr5 | SFI |
| 2 | 1.81±0.03a | 1.74±0.05a | 1.70±0.04a | 1.78±0.01a | 0.5 | 1.02±0.08a | 0.97±0.02a | 1.38±0.06a | 1.73±0.01a |
| 3 | 1.67±0.05b | 1.63±0.02b | 1.59±0.07ab | 1.72±0.01b | 1 | 0.90±0.01ab | 0.87±0.00b | 1.11±0.01b | 1.63±0.01b |
| 4 | 1.56±0.00c | 1.51±0.02c | 1.38±0.00b | 1.58±0.00c | 1.5 | 0.80±0.02b | 0.72±0.01c | 0.94±0.01c | 1.12±0.01c |

AAUPi3 = Addis Ababa University pineapple, AAUOr7 = Addis Ababa University orange, AAUGr5 = Addis Ababa University grape, and SFI = commercial yeast. Within a column, means that are followed by the same letters do not differ significantly (p < 0.05%).

**Table 3. Comparison of the leavening ability of selected yeast isolates and commercial baker's yeast both at room temperature and at 30 °C with different incubation times.**

| Yeast Isolates | Volume increment (mL) at 30 °C | | | | | Volume increment (mL) at room temperature | | | | |
|---|---|---|---|---|---|---|---|---|---|---|
| | 0 hrs | 2 hrs | 4 hrs | 6 hrs | 8 hrs | 0 hrs | 2 hrs | 4 hrs | 6 hrs | 8 hrs |
| AAUGr5+AAUPi3+ AAUOr7 | 60±00a | 112±2.00a | 121±4.73a | 133±3.51a | 195±3.51a | 60±00a | 103±3.00a | 109±2.52a | 123±4.00a | 175±2.52a |
| AAUGr5+AAUPi3 | 60±00a | 97±2.00b | 103±1.53b | 109±2.52b | 152±2.08b | 60±00a | 93±3.00b | 99±2.00b | 106±3.06b | 147±1.00b |
| AAUGr5+AAUOr7 | 60±00a | 89±2.00c | 94±1.00c | 101±2.00c | 144±3.21c | 60±00a | 87±2.00bc | 92±1.53c | 101±3.00bc | 139±2.08c |
| AAUPi3+AAUOr7 | 60±00a | 86±3.00 cd | 92±1.52c | 98±2.01 cd | 137±1.53 cd | 60±00a | 83±2.52 cd | 90±2.00 cd | 96±2.00 cd | 130±2.52d |
| AAUGr5 | 60±00a | 81±2.52de | 88±2.52 cd | 95±2.12cde | 132±3.00d | 60±00a | 78±2.08de | 85±2.52de | 92±2.52de | 122±2.52e |
| SFI | 60±00a | 77±2.00ef | 84±2.50de | 92±2.00def | 121±2.08e | 60±00a | 73±2.00ef | 81±2.00ef | 88±2.52ef | 112±2.00f |
| AAUOr7 | 60±00a | 74±2.00f | 84±3.00de | 89±2.51de | 113±2.08f | 60±00a | 69±2.52f | 77±3.00f | 85±2.00ef | 104±2.52g |
| AAUPi3 | 60±00a | 72±3.00f | 79±2.00e | 86±2.50e | 104±3.51g | 60±00a | 67±2.52f | 76±3.00f | 84±2.52f | 93±2.08h |
| without yeast | 60±00a | 60±0.00g | 60±0.00f | 60±0.00f | 60±0.00h | 60±00a | 60±0.00g | 60±0.00g | 60±0.00g | 60±0.00i |

AAUGr5=Addis Ababa University grape, AAUOr7=Addis Ababa University orange, SFI=commercial yeast, and AAUPi3=Addis Ababa University pineapple. The means in a column that are indicated by the same letters are not statistically different ($p < 0.05\%$).

### 3.6. Leavening ability

The dough-leavening ability of the selected wild yeast isolates was tested both individually and in co-culture at room temperature and 30 °C (Table 3). The yeast isolates were found to raise dough volume better when incubated at 30 °C than at room temperature. The order of the yeast isolates based on their dough leavening ability is as follows: co-culture of the three isolates, AAUGr5+AAUPi3, AAUGr5+AAUOr7, AAUPi3+AAUOr7), AAUGr5, commercial yeast (SFI), AAUOr7, and AAUPi3 (Table 3).

## 4. Discussion

This study's findings could significantly impact the baking industry in Ethiopia by introducing locally sourced wild yeast strains as a sustainable alternative to commercial yeasts. This approach can enhance Ethiopian bread's quality and flavor profile, such as difo dabo or ambasha, while reducing dependency on imported yeast. Furthermore, the use of wild yeasts could lower production costs, benefiting small-scale and local bakeries. However, the study was limited to the use of molecular sequencing for more accurate identification of the yeast isolates at the molecular level and scaling up for industrial use.

This study was carried out to assess the leavening ability of wild yeasts recovered from different fruits and identify them using cultural, morphological, and biochemical tests. Three wild yeast isolates were selected based on their high $CO_2$ production ability and non-$H_2S$ production character. Based on their colony characteristics, the isolates were white and creamy white with smooth or rough margins. Under the microscope, shown to be unicellular and had spherical or oval cell morphologies, and their budding pattern was similar to the commercial yeast isolate. A similar finding was reported by [20] that yeast isolates isolated from pineapple and orange peel belonged to *saccharomyces* type, unicellular ascomycete.

Out of the 88 wild yeast isolates recovered in this study, AAUPi3, AAUOr7, and AAUGr5 were found to be the best $CO_2$ producers as observed in the Durham tube. The yeast isolates, including the commercial yeast isolate, fermented all six sugars tested except lactose and produced higher amounts of $CO_2$ gas. According to earlier findings, *saccharomyces*-type unicellular ascomycetes were tested for their ability to ferment carbohydrates. Pineapple isolate was found to be able to use six of the seven sugars tested, while orange isolate was only able to use five, suggesting that they had different sugar utilization abilities [20].

The ability of *Saccharomyces cerevisiae* to ferment glucose, fructose, galactose, sucrose, and raffinose but not lactose supports its identification [21]. The gas produced, $CO_2$, is crucial in bakeries and the manufacturing of bread [22]. This

could indicate invertase activity exhibited by the yeast isolates, including those recovered and evaluated in this study. Likewise, [23] reported that *Saccharomyces,* incapable of fermenting lactose, lacked a β-galactosidase or lactase system. The commercial yeast isolates and the three wild yeast isolates, when cultured on a medium with $(NH_4)_2SO_4$ as the only nitrogen source, encouraged the growth of the isolates. On the contrary, $KNO_3$ was shown to inhibit the growth of both the commercial and wild yeast isolates. Consequently, nitrogen sources such as aqueous ammonia and ammonium salts should usually be added to maintain the proliferation of yeast cells [24].

The selected wild yeast isolates used in this investigation did not produce $H_2S$. However, the commercial yeast isolate produced a black color in BSA and KIA media that significantly compromised bread quality. $H_2S$ is an undesirable byproduct associated with off-flavors and imparts an unpleasant taste in processed foods, including bread. When baking bread, yeasts that produce a lot of $H_2S$ are not the best choice since they give the bread an unfavorable taste and flavor that lowers its quality [25]. According to another study, a darker color on Lead Acetate Agar (LAA) denotes a higher level of $H_2S$ synthesis [26]. Therefore, given their low $H_2S$ production and strong fermentation ability, the wild yeast isolates selected during this study may be a good option for preparing bread.

In the current study, maximum biomass was obtained after 96 hrs of incubation, and the biomass was shown to decrease with further incubation. Different studies indicated that at the stationary phase growth of yeasts decreases as the rate of metabolism slows down and cell division is halted as a result of high temperatures, poisonous metabolites, and food shortages, which causes cells to autolyze and die [27]. This study's findings are at odds with those of [28], who reported that the maximum biomass was reached after 144 hrs of incubation. The genetic makeup of their cells and the environment under which they were grown may have contributed to the variation.

In the present investigation, the optimal biomass of the selected yeast isolates was recorded at 30 °C. The result of this study is consistent with the findings of other investigators who obtained maximum cell mass production at 30 °C [19]. There was a rapid decrease in cell number for all yeast isolates after 96 hrs of incubation at all tested temperatures since the synthesis of the enzyme was affected by increasing temperature beyond 30 °C [29]. Comparable to this result [30] discovered that the optimum conditions for yeast growth were 30 °C and 72 hrs of incubation.

The findings of this research showed that pH 5 was the optimum value for the growth of yeast isolates, including commercial yeast isolates. In a similar study, *Saccharomyces* was found to grow optimally at a pH of 5 [25]. In general, the selected wild yeast isolates (of this study) were found to perform close to the commercial yeast isolate to pH tolerance, verifying the potential of these wild yeast isolates for leavening of bread dough. Corresponding to this outcome, [30] discovered that yeasts flourished at pH 5 to 5.5 and after 72 hrs of incubation.

The potential wild yeast isolates of this study were able to grow in YEP broth with different concentrations of D-glucose and NaCl. As a result, 2% D-glucose and 0.5% NaCl allowed the best growth of the yeast isolates. These results were found in agreement with previous research, showing that the majority of *Saccharomyces* species isolated from products of traditional fermentation were physiologically acclimated to harsh environments [31].

The current findings showed that the screened co-culture and the individual AAUGr5 wild yeast isolates without $H_2S$ had a superior leavening ability than the commercial yeast isolate. This has demonstrated the combined effects of yeast isolates and the effectiveness of yeast consortia in the leavening of dough [32]. It has been suggested by several researchers [33] that a combination of yeasts (non-*Saccharomyces cerevisiae* + *Saccharomyces cerevisiae*) is essential for high-quality bread leavening and baking purposes, which could explain why the synergy of yeast isolates performed better.

## 5. Conclusion

The findings of this study revealed that a variety of environmental resources, including fruits, may contain potent baker's yeasts that may have a greater ability to leaven dough. For the selected yeast isolates of this study, the optimum growth conditions for yeasts were a temperature of 30 °C, pH 5, and 96 hrs of incubation. Apart from producing a commendable

quantity of $CO_2$, these yeast isolates were non-$H_2S$ producers, in contrast to the commercial yeast isolate that produced $H_2S$. On an individual basis, isolate AAUGr5 had better leavening ability over the commercial yeast isolates. Similarly, the co-culture yeast isolates demonstrated a better leavening effect and a greater potential for industrial application. To optimize the potential use of these isolates at the commercial level, molecular-level identification of the wild yeast strains and cloning of the desired gene to enhance their leavening potential must be conducted. Moreover, pilot-scale trial bread production is recommended.

## Supporting information

**S1 File. D-glucose Results.**
(XLSX)

**S2 File. NaCl Results.**
(XLSX)

**S3 File. pH Result.**
(XLSX)

**S4 File. Temperature Results.**
(XLSX)

**S5 File. Volume at 30 °C Results.**
(XLSX)

**S5 File. Volume at room temp Results.**
(XLSX)

## Acknowledgments

Special thanks to Addis Ababa University/Institute of Biotechnology for allowing us to carry out the study.

## Author contributions

**Conceptualization:** Eshet Lakew Tesfaye, Anteneh Tesfaye Tefera, Diriba Muleta.

**Data curation:** Eshet Lakew Tesfaye, Anteneh Tesfaye Tefera, Diriba Muleta.

**Formal analysis:** Eshet Lakew Tesfaye, Anteneh Tesfaye Tefera, Diriba Muleta.

**Funding acquisition:** Eshet Lakew Tesfaye, Anteneh Tesfaye Tefera, Diriba Muleta.

**Investigation:** Eshet Lakew Tesfaye, Anteneh Tesfaye Tefera, Diriba Muleta.

**Methodology:** Eshet Lakew Tesfaye, Anteneh Tesfaye Tefera, Diriba Muleta.

**Project administration:** Eshet Lakew Tesfaye, Anteneh Tesfaye Tefera, Diriba Muleta.

**Resources:** Eshet Lakew Tesfaye, Anteneh Tesfaye Tefera, Diriba Muleta.

**Software:** Eshet Lakew Tesfaye, Anteneh Tesfaye Tefera, Diriba Muleta.

**Supervision:** Eshet Lakew Tesfaye, Anteneh Tesfaye Tefera, Diriba Muleta.

**Validation:** Eshet Lakew Tesfaye, Anteneh Tesfaye Tefera, Diriba Muleta.

**Visualization:** Eshet Lakew Tesfaye, Anteneh Tesfaye Tefera, Diriba Muleta.

**Writing – original draft:** Eshet Lakew Tesfaye, Anteneh Tesfaye Tefera, Diriba Muleta.

**Writing – review & editing:** Eshet Lakew Tesfaye, Anteneh Tesfaye Tefera, Diriba Muleta.

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
