## [Decision Letter · Decision Letter 0]

22 Jan 2025

PONE-D-24-56564Assessing the Bread-Leavening Ability of Wild Yeasts Isolated from Selected Fruits Collected from Local MarketsPLOS ONE

Dear Dr. Tesfaye,

Thank you for submitting your manuscript to PLOS ONE. After careful consideration, we feel that it has merit but does not fully meet PLOS ONE’s publication criteria as it currently stands. Therefore, we invite you to submit a revised version of the manuscript that addresses the points raised during the review process.

We look forward to receiving your revised manuscript.

Kind regards,

Samuel Adelani Babarinde, PhD

Academic Editor

PLOS ONE

Additional Editor Comments (if provided):

Reviewers' comments:

Reviewer's Responses to Questions

**Comments to the Author**

1. Is the manuscript technically sound, and do the data support the conclusions?

Reviewer #1: Yes

Reviewer #2: Yes

2. Has the statistical analysis been performed appropriately and rigorously? 

Reviewer #1: Yes

Reviewer #2: Yes

3. Have the authors made all data underlying the findings in their manuscript fully available?

Reviewer #1: Yes

Reviewer #2: Yes

4. Is the manuscript presented in an intelligible fashion and written in standard English?

Reviewer #1: Yes

Reviewer #2: Yes

5. Review Comments to the Author

Reviewer #1: This manuscript focuses on evaluating the bread fermentation ability of wild yeasts isolated from specific fruits. It excels in several areas but also has room for improvement.

The motivation for the study is reasonable and sustainable. In Ethiopia, the baking industry heavily relies on imported yeast and baking powder, which is costly, providing a clear incentive to explore alternative sources of yeast. Additionally, the issue of hydrogen sulfide production by commercial yeasts, which negatively affects bread quality, makes the search for more suitable yeast strains particularly necessary. The research objectives are well-defined and closely related to these practical issues, aiming to identify efficient wild yeast strains that could improve the bread-making process.

The experimental design is technically sound and carefully planned. Various fruit samples, including avocado, banana, grape, mango, orange, papaya, and pineapple, were collected from local markets. The isolation of wild yeasts followed a standard procedure, where yeast cells were obtained by serial dilution and cultured on potato dextrose agar (PDA) plates containing chloramphenicol. The selection process based on CO₂ production and the absence of hydrogen sulfide is a reasonable approach for identifying potentially useful yeast strains, aligning well with the study’s objectives. Subsequent identification steps, including culturing, morphological, and biochemical analyses, provide a comprehensive understanding of the selected yeast strains. Using the Dutch DSM baking strain as a standard reference is a common and reliable practice, and previous studies have also used it as a control, so its credibility is likely not in question. The use of SPSS for data analysis is also appropriate.

However, there are some areas for improvement. In the methodology section, although the experimental steps are described in detail, some of the references for material processing methods are somewhat outdated. Although my research focus is not on yeast extraction, in the rapidly evolving field of microbiology, the authors should clarify whether there are more suitable processing methods or whether these methods are considered the gold standard. This would enhance the accuracy and efficiency of the research process.

Another issue is data availability. Given the promising results presented in the manuscript, the authors could point out appropriate ways to access the raw data.

Finally, the manuscript spends a considerable amount of space on the selection, identification, and characterization of yeast strains. However, in evaluating fermentation ability, focusing only on the dough's volume change seems somewhat limited. Bread quality is a multidimensional attribute, and other aspects such as texture, color, and flavor also play a crucial role in determining its overall acceptability. These aspects might also be suitable for assessing the fermentation ability of yeast strains, providing a more comprehensive evaluation of their performance. For example, including images from different stages of fermentation could provide valuable visual information and enhance the clarity of the results.

While the study shows potential, these possible flaws should be addressed. I believe this research will not only contribute to the field of yeast studies but could also impact the baking industry in Ethiopia and other regions. Overall, it is a good study.

Reviewer #2: This study effectively explores the potential of wild yeast isolates from fruits for dough leavening, showing promising results in CO2 production and non-H2S generation. The research is well-structured, with thorough biochemical, cultural, and morphological analysis, and highlights the superiority of co-culture systems for industrial application. However, Minor revision is required to make the manuscript more effective

6. PLOS authors have the option to publish the peer review history of their article (what does this mean? ). If published, this will include your full peer review and any attached files.

**Do you want your identity to be public for this peer review?** For information about this choice, including consent withdrawal, please see our Privacy Policy .

Reviewer #1: No

Reviewer #2: **Yes: ** Dr. Muhammad Sibt-e-Abbas

---

## [Author Response · Author response to Decision Letter 0]

28 Jan 2025

We tried to address all issues and concerns raised by the academic editor and the reviewers.

---

## [Decision Letter · Decision Letter 1]

24 Mar 2025

PONE-D-24-56564R1Assessing the Bread-Leavening Ability of Wild Yeasts Isolated from Selected Fruits Collected from Local MarketsPLOS ONE

Dear Dr. Tesfaye,

Thank you for submitting your manuscript to PLOS ONE. After careful consideration, we feel that it has merit but does not fully meet PLOS ONE’s publication criteria as it currently stands. Therefore, we invite you to submit a revised version of the manuscript that addresses the points raised during the review process.

Please submit your revised manuscript by May 08 2025 11:59PM.  If you will need more time than this to complete your revisions, please reply to this message or contact the journal office at plosone@plos.org . Please include the following items when submitting your revised manuscript:

We look forward to receiving your revised manuscript.

Kind regards,

Samuel Adelani Babarinde, PhD

Academic Editor

PLOS ONE

Journal Requirements:

Reviewers' comments:

Reviewer's Responses to Questions

**Comments to the Author**

1. If the authors have adequately addressed your comments raised in a previous round of review and you feel that this manuscript is now acceptable for publication, you may indicate that here to bypass the “Comments to the Author” section, enter your conflict of interest statement in the “Confidential to Editor” section, and submit your "Accept" recommendation.

Reviewer #1: All comments have been addressed

Reviewer #3: (No Response)

2. Is the manuscript technically sound, and do the data support the conclusions?

Reviewer #1: Yes

Reviewer #3: (No Response)

3. Has the statistical analysis been performed appropriately and rigorously? 

Reviewer #1: Yes

Reviewer #3: (No Response)

4. Have the authors made all data underlying the findings in their manuscript fully available?

Reviewer #1: Yes

Reviewer #3: (No Response)

5. Is the manuscript presented in an intelligible fashion and written in standard English?

Reviewer #1: Yes

Reviewer #3: (No Response)

6. Review Comments to the Author

Reviewer #1: The authors have made remarkable improvements to the manuscript. The structural optimization has significantly enhanced the overall readability and logical coherence, guiding readers more smoothly through the research process from start to finish. By eliminating potentially misleading expressions, the clarity of the content has been greatly improved, ensuring that the scientific information is accurately conveyed. The expanded discussion on the impact on Ethiopian agriculture adds a practical and far - reaching dimension to the research, demonstrating the potential real - world implications of the study. Although some issues raised in the previous review remain unaddressed, likely due to the current research scale, this does not detract from the article's innovation and advantages. The exploration of wild yeasts for bread - leavening is novel and holds great promise. Overall, the article now substantially meets the basic requirements of PLOS One for published manuscripts, and it is expected that the authors may further address these issues in future research endeavors.

Reviewer #3: Kindly recheck language and grammar issues to improve clarity, coherence, and overall professionalism of the manuscript.

7. PLOS authors have the option to publish the peer review history of their article (what does this mean? ). If published, this will include your full peer review and any attached files.

**Do you want your identity to be public for this peer review?** For information about this choice, including consent withdrawal, please see our Privacy Policy .

Reviewer #1: No

Reviewer #3: No

---

## [Author Response · Author response to Decision Letter 1]

27 Mar 2025

I have attached documents to respond to reviewers comments.

---

## [Decision Letter · Decision Letter 2]

23 Apr 2025

Assessing the Bread-Leavening Ability of Wild Yeasts Isolated from Selected Fruits Collected from Local Markets

PONE-D-24-56564R2

Dear Dr. Tesfaye,

We’re pleased to inform you that your manuscript has been judged scientifically suitable for publication and will be formally accepted for publication once it meets all outstanding technical requirements.

Kind regards,

Samuel Adelani Babarinde, PhD

Academic Editor

PLOS ONE

Additional Editor Comments (optional):

Reviewers' comments:

Reviewer's Responses to Questions

**Comments to the Author**

1. If the authors have adequately addressed your comments raised in a previous round of review and you feel that this manuscript is now acceptable for publication, you may indicate that here to bypass the “Comments to the Author” section, enter your conflict of interest statement in the “Confidential to Editor” section, and submit your "Accept" recommendation.

Reviewer #3: All comments have been addressed

Reviewer #4: All comments have been addressed

2. Is the manuscript technically sound, and do the data support the conclusions?

Reviewer #3: Yes

Reviewer #4: Yes

3. Has the statistical analysis been performed appropriately and rigorously? 

Reviewer #3: (No Response)

Reviewer #4: Yes

4. Have the authors made all data underlying the findings in their manuscript fully available?

Reviewer #3: (No Response)

Reviewer #4: Yes

5. Is the manuscript presented in an intelligible fashion and written in standard English?

Reviewer #3: (No Response)

Reviewer #4: Yes

6. Review Comments to the Author

Reviewer #3: Substantial revision was done by authors and in present form, the manuscript in revised form can be accepted

Reviewer #4: (No Response)

7. PLOS authors have the option to publish the peer review history of their article (what does this mean? ). If published, this will include your full peer review and any attached files.

**Do you want your identity to be public for this peer review?** For information about this choice, including consent withdrawal, please see our Privacy Policy .

Reviewer #3: No

Reviewer #4: **Yes: ** Alaa Jabbar ABD Al-Manhel

---

## [Editor Report · Acceptance letter]

PONE-D-24-56564R2

PLOS ONE

Dear Dr. Tesfaye,

I'm pleased to inform you that your manuscript has been deemed suitable for publication in PLOS ONE. Congratulations! Your manuscript is now being handed over to our production team.

Kind regards,

on behalf of

Dr. Samuel Adelani Babarinde

Academic Editor

PLOS ONE